# The T6SS-Dependent Effector Re78 of *Rhizobium etli* Mim1 Benefits Bacterial Competition

**DOI:** 10.3390/biology12050678

**Published:** 2023-05-04

**Authors:** Bruna Fernanda Silva De Sousa, Lucía Domingo-Serrano, Alvaro Salinero-Lanzarote, José Manuel Palacios, Luis Rey

**Affiliations:** 1Centro de Biotecnología y Genómica de Plantas, Instituto Nacional de Investigación y Tecnología Agraria y Alimentaria (INIA/CSIC), Campus de Montegancedo UPM, Universidad Politécnica de Madrid (UPM), 28223 Pozuelo de Alarcón, Spain; brunasousa.agronomia@gmail.com (B.F.S.D.S.);; 2Departamento de Biotecnología y Biología Vegetal, ETSI Agronómica Alimentaria y de Biosistemas, Universidad Politécnica de Madrid, 28040 Madrid, Spain

**Keywords:** type six secretion, new effector–immunity pair, *Rhizobium*–legume symbiosis, bacterial competition, nodule, T6SS

## Abstract

**Simple Summary:**

*Rhizobium etli* Mim1 (ReMim1) possesses a protein secretion system type VI (T6SS) that is active in free living and symbiosis. The T6SS is a nanosyringe that secretes proteins called effectors to both eukaryotic and prokaryotic target cells. The ReMim1 T6SS gene cluster encodes for a toxic effector (Re78) together with an immunity protein (Re79) as demonstrated when expressed in *Escherichia coli*. In addition, it was observed that the toxic role of the Re78 protein is outside the cytoplasm since its toxic effect on *E. coli* only occurred when a signal peptide was added to it. Re79 is found in ReMim1 periplasm and is T6SS-independent for its translocation. Additionally, the Re78/Re79 pair is involved in bacterial competition and nodule occupancy. A better understanding of the role of this secretion system can be very useful to select highly competitive rhizobia for inoculants.

**Abstract:**

The genes of the type VI secretion system (T6SS) from *Rhizobium etli* Mim1 (ReMim1) that contain possible effectors can be divided into three modules. The mutants in them indicated that they are not required for effective nodulation with beans. To analyze T6SS expression, a putative promoter region between the *tssA* and *tssH* genes was fused in both orientations to a reporter gene. Both fusions are expressed more in free living than in symbiosis. When the module-specific genes were studied using RT-qPCR, a low expression was observed in free living and in symbiosis, which was clearly lower than the structural genes. The secretion of Re78 protein from the T6SS gene cluster was dependent on the presence of an active T6SS. Furthermore, the expression of Re78 and Re79 proteins in *E. coli* without the ReMim1 nanosyringe revealed that these proteins behave as a toxic effector/immunity protein pair (E/I). The harmful action of Re78, whose mechanism is still unknown, would take place in the periplasmic space of the target cell. The deletion of this ReMim1 E/I pair resulted in reduced competitiveness for bean nodule occupancy and in lower survival in the presence of the wild-type strain.

## 1. Introduction

Rhizobia are α and β proteobacteria capable of fixing nitrogen in symbiosis with legumes [1]. Among the rhizobial genes described that are important for symbiosis are those that encode for protein secretion systems and can translocate proteins called effectors to target cells [2,3,4,5].

One of them is the Type VI Secretion System (T6SS) [6]. This system is a multiprotein nanosyringe employed by many Gram-negative bacteria to translocate effectors in neighboring cells, both prokaryotic and eukaryotic, or to the extracellular milieu. The functions of effectors are diverse but the most frequently described of them is to act against competitor bacteria (lipases, DNases, peptidoglycanases, etc.); they have also been involved in eukaryote pathogenesis and in the uptake of metals [7,8].

The operation and formation of a T6SS nanosyringe typically requires 13 conserved proteins referred to as TssA to TssM. The nanosyringe contains an inner tube of TssD (also named Hcp) surrounded by a contractile outer sheath made of TssBC and topped by a spike of TssI (VgrG) and PAAR protein. The TssA anchors to the membrane and allows for the formation of the sheath that is docked at the other end to a TssEFGK baseplate attached to the TssJLM transmembrane complex. After contraction, the sheath is recycled by the ATPase TssH (ClpV) [6,9,10,11].

The TssA-M and conserved accessory proteins (Tag) are generally encoded by a gene cluster, often comprising 20–30 genes, which also include strain-specific effectors and adaptor proteins [7]. The genes encoding antibacterial effectors are often adjacent to the genes encoding the so-called immunity proteins that prevent autointoxication or intoxication produced by sister cells [12,13].

T6SS genes are present in 25% of proteobacterial genomes, including plant-associated bacteria and several rhizobial species [14,15,16]. T6SS presents an important role in interbacterial competition and provides advantages in multimicrobial plant environments [15]. In most cases, the T6SSs of the plant microbiota act against microbial competitors, the host plant, or both, depending on the repertoire of effectors carried by each strain [17]. T6SS-dependent effectors that are capable of directly affecting plant cells have not yet been identified.

The role of rhizobial T6SS in symbiosis with legumes has hardly been studied. The first description of the role of T6SS was made on a set of genes that impaired pea nodulation in *Rhizobium leguminosarum* RBL5523 [18]. Later, it was observed that the presence of T6SS does not affect the symbiotic effectiveness of *Paraburkholderia phymatum*/*Vigna unguiculata* [19] and *Azorhizobium caulinodans*/*Sesbania rostrata* [20]. In these cases, the role of T6SS in interbacterial and symbiotic competitiveness was demonstrated. Our group recently found the positive role of T6SS on the symbioses of both *Rhizobium etli*/*Phaseolus vulgaris* and *Bradyrhizobium* sp.*/Lupinus angustifolius*; in both cases, mutants in the structural genes of the T6SSs induced plants with up to 40% less shoot dry weight than wild-type, and up to 60% less nodule fresh weight [21,22].

*R. etli* Mim1 (ReMim1) encodes a T6SS that, phylogenetically, belongs to the highly homogeneous group 5, which corresponds mainly to *Rhizobium* strains [15]. Three ReMim1 mutants, affected in the T6SS structural genes (hcp::pk18, ΔtssM, and ΔtssA-tagE), induced bean plants with lower shoot dry weight and smaller nodules than the wild-type strain [21]. This work aims to identify and characterize the effectors encoded in the ReMim1 T6SS gene cluster and to study their possible involvement in symbiosis. We provide evidence supporting the expression of the system in free-living conditions and in symbiosis with beans via the transcriptional fusions of the T6SS promoter region and using RTq-PCR. We also show an in silico functional characterization of the putative effectors and demonstrate the role of a novel effector/immunity pair (E/I) in bacterial competition.

## 2. Materials and Methods

### 2.1. Bacterial Strains, Plasmids, and Growth Conditions

The strains and plasmids used in this study can be found in Appendix A. *E. coli* strain DH5α was used for general cloning, *E. coli* strain BL21 (DE3) was used as the host for protein expression, and *E. coli* S17.1 was used for plasmid conjugation. The *E. coli* strains were cultured at 37 °C and were grown in Luria–Bertani (LB) medium [23]. The *Rhizobium* strains were routinely cultured at 28 °C in Yeast Mannitol Broth (YMB) [24], Tryptone Yeast (TY) [25], and Universal Minimal Salt medium (UMS) [26], whereas the selection of the transconjugants was carried out in *Rhizobium* minimal broth (Rmin) [27]. The OD_600_ values were measured in the Bioscreen C° Pro (Oy Growth Curves Ab Ltd., Helsinki, Finland). The media were supplemented with the following antibiotics as required in μg/mL: ampicillin, 100 (*E. coli*), chloramphenicol, 30 (*E. coli*), kanamycin, 50 (*Rhizobium*), tetracycline, 5 (*E. coli*, *Rhizobium*), gentamicin, 20 (*Rhizobium*), and spectinomycin, 50 (*Rhizobium*).

### 2.2. Construction of T6SS Mutants

#### 2.2.1. Generation of Mutants ∆re78–79 and ∆re84–89

The generation of the ∆re78–79 mutant was performed using fusion PCR. Two fragments flanking the region to be deleted were amplified with the primers included in Appendix A. These fragments were fused with another PCR using the end primers and taking advantage of the fact that the internal primers have overlapping sequences. The fusion amplicon was cloned into the PCR2.1 TOPO vector (Invitrogen, Paisley, UK). Then, it was extracted using *Eco*RI (New England Biolabs, Hitchin, UK) digestion according to standard protocols [23] and cloned into plasmid pk18mobsacB. *E. coli* S17.1. The competent cells were transformed with the new plasmid and were used as a donor strain to conjugate ReMim1. The single recombinants were selected in Rmin kanamycin and the second recombination was favored when growing the cells without kanamycin in Rmin 10% sucrose. A similar process was followed for obtaining the deletion mutant ∆re84–89.

#### 2.2.2. Mutant re82HD-AA

The *re82* gene was amplified using P82.F and P82.R and then cloned into the pSCA vector (Appendix A). From the pSCA (Agilent Technologies, Santa Clara, CA, USA) derivative, another PCR was performed with two complementary primers comprising the sequence coding for an Alanine that will replace H_431_. Once the sequence was verified, another PCR was performed with two other primers comprising the sequence coding for an Alanine that will replace D_434_. The mutation transfer to ReMim1 was from the conjugation of a pk18mobsacB derivative according to Section 2.2.1.

### 2.3. Plasmid Constructions to Express re78 and re79 in E. coli and ReMim1

The genes *re*78 and *re*79 were amplified independently using PCR from ReMim1 total DNA. The *re*78 was cloned into pET22b in two ways, one incorporating the PelB signal peptide sequence (PelB) from the vector into the gene (the PelB leader peptide attached to a protein, directing it to the *E. coli* periplasm) and the other without the PelB sequence. For this purpose, two different primer pairs and *Bam*HI/*Xho*I and *Nde*I/ *Xho*I restriction enzymes, respectively, were used. The *re*79 gene was cloned into pBAD33 after PCR amplification and digestion with *Xba*I and *Hind*III. The restriction enzymes are from New England Biolabs, UK.

To identify Re79 in ReMim1, the *re79* gene was amplified using PCR with a primer with a sequence extension to obtain Re79 with a C-terminal Strep-tag (Appendix A). The amplicon was cloned in a PCR2.1 TOPO plasmid. The insert was then extracted using *Nde*I restriction enzyme and cloned in pLMB509 plasmid. *E. coli* S17.1 was transformed with plasmid pLMB509-Re79StrepTag and conjugated to ReMim1 and hcp::pk18 strains according to 2.2.1. item. All the primers used are listed in Appendix A. All the constructs were confirmed using Sanger sequencing.

### 2.4. E. coli Toxicity Assay

To evaluate the toxicity of putative effectors, the *E. coli* BL21(DE3) cells were transformed with plasmids expressing Re78 (with and without PelB) and/or Re79. The cells were grown in LB medium to the exponential phase, then washed with 0.9% NaCl. Each culture was standardized to OD_600_ = 0.01 in 200 μL of LB medium, which was transferred to 100-well plates (Honey plates) in quadruplicates with the appropriate antibiotics and containing inductors, 1 mM isopropyl 1-thio-β-D-galactopyranoside (IPTG) (ITW Reagents PanReac AppliChem, Castellar del Vallès, Spain), and 0.2% L-arabinose (*wt./vol.*) (Sigma-Aldrich, St. Louis, MO, USA) or the repressor D-glucose 0.2% (*wt./vol.*) (Fisher Chemical, Loughborough, UK). The plates were incubated for 7 h with orbital agitation at 37 °C, and the OD_600_ values were measured every 30 min in the Bioscreen C° Pro (Oy Growth Curves Ab Ltd., Helsinki, Finland).

### 2.5. E. coli Staining for Microscope Viewing

The *E. coli* staining was performed with cells collected after 3 h growing with 1 mM of IPTG; 150 μL of the culture was centrifuged at 8000× *g* for 10 min, and the cells were resuspended in 15 μL of medium then heat-fixed on a slide and stained with 1% crystal violet (*wt./vol.*) (Thermo Fisher Scientific, Bremen, Germany) during 30 s and washed three times with distilled water. The samples were visualized using Leica DM2000 microscope and a Leica DFC 300FX camera (1.4 Mpixels) at 100× resolution and analyzed using ImageJ [28].

### 2.6. Subcellular Fractionation and Immunodetection of Re79 Protein from Rhizobium

The *rhizobium* cells harboring pLMB509-Re79StrepTag grown at 28 °C overnight with 3 mM of taurine were harvested using centrifugation (15 min, 8000× *g*, 4 °C). The subcellular fractionation protocol was adapted from Sibinelli–Sousa and collaborators based on osmotic shock to obtain the periplasmic fraction, sonication, and ultracentrifugation to obtain the cytoplasmic and membrane fractions [29]. The protein extracts were separated by SDS-PAGE 12% of acrylamide and transferred to a Polyvinylidene fluoride 0.45 micron filter (InmobilonTM-P, Millipore, Burlington, MA, USA). The proteins were detected using Strep-Tactin conjugated to alkaline phosphatase antibody (1:2500; IBA, Munich, Germany) and the immunoblots were developed using a chromogenic substrate (bromochloroindolyl phosphate-nitroblue tetrazolium) according to the procedure of the manufacturer (Bio-Rad Laboratories, Inc., Hercules, CA, USA). The protein sizes were estimated by comparing their migration rates with those of a reference standard (BlueStar Prestained Protein Ladder Nippon Genetics, Cultek, Madrid, Spain). The theoretical molecular mass value of Re79StrepTag was estimated using the ProtParam web (http://web.expasy.org/protparam/, accessed on 27 August 2020).

### 2.7. Plant Assays

The *Phaseolus vulgaris* cv Negro Jamapa seeds were used in the plant assays. The seeds were surface sterilized (1 min in ethanol absolute, 3 min in bleach 12% *wt./vol.*, and 10 washes of sterile distilled water) and axenically germinated on 1% agar plates at 20 °C in the dark to germinate for 3 days. The seedlings were transferred into sterilized Leonard jars containing sterilized vermiculite, and nitrogen-free Jensen’s solution [24]. The *Rhizobium* cell growth in YMB medium (1 mL, 10^8^–10^9^ cells/mL) were used to inoculate the seedlings. The non-inoculated jars were used as negative nodulation controls. Each jar contained two plants that were grown under bacteriologically controlled conditions for 28 days post-inoculation (dpi). The plants were grown in a greenhouse adjusted to 18/25 °C (night/day) temperatures and 16/8 h light/dark photoperiod. The fresh nodules were counted and weighed. The shoot and nodules dry weights were determined after drying at 60 °C for 72 h (adapted from [21]). The presented values are the mean of at least 6 plants from 4 biological replicates.

### 2.8. Bacterial Competition Assay

#### 2.8.1. Competitiveness in Free Living

The bacterial competition on agar plates was performed by a co-culture of wild-type (wt) strain ReMim1 and mutants. The cells were grown in TY medium to OD_600_ = 0.6, which are conditions in which ReMim1 Hcp expression has been previously demonstrated [21], and then washed with 0.9% NaCl and adjusted to an OD_600_ of 0.1. The mixtures of 20 μL with 1:1 ratio attacker:prey were spotted in triplicate onto 0.2 μm filter membranes that were placed on TY agar. The plates were incubated overnight at 28 °C. The competition outcome was quantified by counting the colonies grown on plates with the corresponding antibiotics, tetracycline for strains carrying the pHC60 plasmid, and spectinomycin for strains carrying pCMB13.

#### 2.8.2. Competitiveness in Symbiosis

Different ratios (1:10, 1:1, and 10:1) of ReMim1-pHC60 and the mutant Δre78–79-pCMB13 were inoculated directly into the pre-germinated Negro Jamapa bean seeds. We calculate the number of cells by determining that 1 OD_600_ = 10^7^ CFU/mL. ReMim1-pHC60 against ReMim1-pCMB13 competition was used as a competition control. The nodules were separated from the roots 28 dpi and sterilized with ethanol absolute during 1 min and bleach 6%, 3 min. After 10 washes with sterile distilled water, the nodules were crushed individually with 0.9% NaCl in 96-well plates. The nodule fluorescence was determined using iBright Image System (Thermo Fisher Scientific, Berlin, Germany). The nodule cells were also cultured on YMB plates where the fluorescence of each strain was determined.

### 2.9. T6SS Expression of ReMim1

#### 2.9.1. Expression of Promoter Region

To analyze the expression of T6SS in strain ReMim1, a 1242-bp region (P6), comprising 557 bp of the *tss*A gene, 455 bp of *tss*H, and 230 bp of the intergenic region between both genes, which likely includes the promoter region, was amplified with primers as shown in Appendix A. P6 was fused in front of the *lac*Z gene of plasmid pMP220 (Appendix A). The plasmids containing transcriptional fusions in both orientations were named pTssA and pHcp, depending on which gene they were oriented toward. The P6 derivatives were conjugated to the ReMim1 strain. The orientation of the promoter region with respect to the *lac*Z gene was determined using PCR. The pTssA and pHcp expression was studied under different free-living conditions and in bean bacteroids. The free-living cells were adjusted to an initial OD_600_ of 0.1 and, after 3 h, the cells were harvested for β-galactosidase activity. The bacteroids were isolated following the protocol of Ruiz-Argüeso [30] after 28 dpi with ReMim1 harbouring pTssA, pHcp, or pMP220 (empty plasmid was used as control). The β-galactosidase activity was measured following the protocol described by Miller [31].

#### 2.9.2. RNA Extraction, cDNA Synthesis, and qRT-PCR of T6SS Genes

The nodules from the *Phaseolus vulgaris* cv Negro Jamapa plants were harvested in liquid nitrogen after 28 dpi inoculated with ReMim1. The free-living samples were grown in TY media at 28 °C, and the cells were harvested in stationary phase. The RNA was obtained using TRI-Reagent (Life Technologies, Renfrew, UK), DNase turbo (Life Technologies, UK), and RNasy Mini Kit (Qiagen, Hilden, Germany). The PrimeScript RT Reagent Kit (Takara, San Jose, CA, USA) supplemented with RNAse out (Invitrogen, London, UK) was used to synthesize cDNA from 500 ng. The T6SS gene expression was carried out using real-time reverse transcription polymerase chain reaction (RT-qPCR) using 1/10 dilution of cDNA with the primers listed in Appendix A and a Power SYBR^®^ Master Mix (Applied Biosystems, Foster City, CA, USA) with an annealing temperature of 58 °C using the LightCycler^®^480II (Roche, Basel, Switzerland). The relative expressions of the target genes were normalized in relation to the housekeeping RNA polymerase sigma factor (*rpoD*). The transcript levels were calculated using the 2^−∆∆Ct^ method [32]. The results are the average of the two biological independent assays with three replicates [32].

### 2.10. Preparation and Analysis of Secretomes

The cells were grown in YMB to the stationary phase, and the supernatant was collected using centrifugation for 1 h at 4000× *g* 4 °C and lyophilized after freezing with liquid nitrogen. The proteins were precipitated with 0.5 M Tris-HCl pH 6.8, 1 M dithiothreitol (DTT), 10% SDS, and phenol. The precipitate was collected using centrifugation (30 min, 4000× *g*, 4 °C); 1 M DTT and 8 M ammonium acetate were added during 30 min. Methanol and 70% of ethanol were used to clean the pellet. The pellet was washed with acetone and centrifuged (10 min, 10,000× *g*, 4 °C). The proteins were resuspended in 10 mM Tris and digested using trypsin (Promega, Madison, WI, USA) [33,34] and analyzed using Reverse phase-liquid chromatography RP-LC-MS/MS [35].

### 2.11. Bioinformatic Analysis

All the sequences identified in this study were from the National Center for Biotechnology Information (NCBI) in the non-redundant (nr) protein data base. The homologous proteins of the genomic context were obtained using BLAST (Basic Local Alignment Search Tool). The presence of signal peptides was investigated using SignalP-5.0 [36]. The InterproScan webserver was used for the domain prediction of the proteins (https://www.ebi.ac.uk/interpro/, accessed on 23 February 2023). The multiple alignments were performed using ClustalW version 1.2.4. of UniprotKB web (https://www.uniprot.org/, accessed on 23 February 2023) and visualized using the Bioedit Sequence Alignment Editor version 7.2.5 [37].

### 2.12. Statistical Analysis

The statistical analyses were performed using GraphPad Prism9 Software (Boston, MA, USA) version 9 of the software used. All the statistical tests, numbers, replicates and biological experiments, and standard deviation (SD) of the mean are reported in the figure legends.

## 3. Results

### 3.1. Bioinformatic Analysis of Genes That Could Encode Effectors in the Re Mim1 T6SS Gene Cluster

Our model bacterium ReMim1 has a T6SS encoded by two divergent clusters of 14 genes. One cluster contains structural and regulatory genes and the other just structural genes, such as *hcp* (*tssD*), *vgrG* (*tssI*), and *tssH*, and genes mostly of unknown function that may encode novel effectors (Figure 1). These genes were grouped into three modules for analysis and comprised *re78* and *re79* in module 1, *re81–re83* in module 2, and *re84–re89* in module 3 (Figure 1a). In a genomic context (Figure 1a), module 1 mostly contains the homologs from rhizobia (*Neorhizobium, Sinorhizobium*, *R. phaseoli*, *R. ruizarguesonis*, *R. oryziradicis*, and *Rhizobium* sp. Leaf262) that are present in most T6SS clusters. The Re78–79 homologous were also found in *Aureimonas pseudogalii* (in a T6SS cluster), in *Azospirillum* sp., and in *Sphingomonas sanguinis* (Appendix A). To obtain evidence about the possible function of the proteins encoded by modules 1–3, a bioinformatics analysis was performed. The analysis (Figure 1b) predicts that the Re79 (REMIM1_PF00479) and Re88 (REMIM1_PF00488) proteins have a signal peptide, thus indicating that their function is likely extracytoplasmic, which is consistent with the presence of non-cytoplasmic domains. This type of domain is also present in Re78 (REMIM1_PF00478), which also has a small cytoplasmic and a transmembrane domain, thus suggesting that it is membrane bound and that most of the sequence is outward oriented.

In module 2, Re81 (REMIM1_PF00481), Re82 (REMIM1_PF00482), and Re83 (REMIM1_PF00483) code for an adaptor and toxic effector and the cognate immunity protein, respectively. Re81 shows the DUF2169 domain identified and characterized by Bondage and collaborators [38]. Re82 possesses an N-terminal PAAR-like domain found at the N-terminal of bacterial toxins as Tse7, which is a DNase secreted by the T6SS in *Pseudomonas aeruginosa* [39], and two novel toxin domains, “Tox15”, the second of which contains the HxxD catalytic motif required for DNase activity [40] (Appendix A). The Re83 protein displays two domains, an N-terminal GAD-related domain and Tde1C at the C-terminal end identified in the T6SS immunity protein Tdi1 from *Agrobacterium tumefaciens* and other bacterial proteins [40]. The first of these domains also appears in the module 3 proteins Re84 (REMIM1_PF00484) and Re86 (REMIM1_PF00486), while the second appears only in Re86. Re85 (REMIM1_PF00485) and Re87 (REMIM1_PF00487) align with a high degree of conservation with larger proteins and may be a truncated version of them. Re85 (71 aa) has 87% conservation to a *Rhizobium* SMI1/KNR4 family protein (190 aa) (WP_040112315) (Appendix A). These SMI1/KNR4 domains are present in some immunity proteins from bacterial-contact-dependent toxin systems [41] and some are within T6SS clusters [42]. Re87 is homologous to a S-adenosyl-L-methionine-dependent methyltransferase protein (REMIM1_CH00796, 340 aa) encoded on the ReMim1 chromosome, that has the SAM-dependent methyltransferase domain partially conserved (Appendix A). REMIM1_CH00796 is not found in a T6SS gene context. Re89 (REMIM1_PF00489) has 73 aa and has no identifiable domains or significant homology to any protein in the databases.

### 3.2. Symbiotic Phenotype of ReMim1 Mutants in Genes That Could Encode T6SS-Dependent-Effectors

In 2019, we demonstrated that mutations in structural genes in the T6SS of ReMim1 induced impaired symbiosis in bean nodules [21]. This led us to analyze the symbiotic role of genes from modules 1–3. Three mutant strains were made to disable the potential effectors in these regions. In two of these mutants, the *re78–re79* (∆re78–79) and *re84–re89* (∆re84–89) are deleted. The third mutant targeted the *re82* gene, which encodes as mentioned a protein with a DNase domain (HxxD). The mutant, designated re82HD-AA, substituted H_431_ and D_434_ with alanines. All the mutants showed similar growth to the wt strain in media such as TY or YMB (Appendix A).

The symbiotic phenotype of the wt and the three mutants with *P. vulgaris* cv. Negro Jamapa was examined at 28 dpi (Figure 2a,b). The results indicated that the shoot dry weight was similar in plants inoculated with the three mutants with respect to the wt strain. However, the number of nodules was higher for re82HD-AA and ∆re84–89 mutants, and the nodule fresh weight was also higher in the plants induced by the ∆re78–79 and ∆re84–89 mutants. The uninoculated plants and the plants inoculated with a non-polar *hcp* mutant (hcp::pk18) were included as controls; the shoot and nodule biomass values in the *hcp* mutant were lower than in the wt, as previously demonstrated [21]. These results indicate that the genes encoding these potential effectors are not responsible for the beneficial effect previously attributed to T6SS on symbiosis.

### 3.3. T6SS Gene Expression

The previous work showed that ReMim1 Hcp was present in both free-living cells and bean bacteroids [21]. To learn more about T6SS gene expression, promoter transcriptional fusions were performed. The DNA region was between the *tssA* and *tssH* genes (P6) of ReMim1, which likely corresponds to a promoter region for the two divergent T6SS gene clusters (Figure 1) and was used to assess their expression in different conditions. P6 was fused in both orientations in front of the reporter gene *lacZ* of the pMP220 vector generating pTssA and pHcp (the *hcp* gene is in the same orientation as *tssH*). The expression of the fusions in ReMim1 was detected both in free living and in bean bacteroids. The values were higher in free living than in symbiosis, and pTssA produced higher activity than pHcp in both conditions (Figure 3a,b).

The RT-qPCR was carried out to further determine the expression of genes from the ReMim1 T6SS modules. The analysis included the structural genes *tssB* and *hcp* and the genes *re78–79* and *re81–89* (Figure 3c), whose expression was normalized using the amplification of the sequence corresponding to the *rpoD*. The level of *tssB* gene expression was lower in the bacteroids than in the free-living cells, whereas the *hcp* gene had similar levels in both conditions. For the rest of the genes, only *re78*, *re79*, and *re86* in symbiosis and *re78* and *re81* in free living showed detectable values, though they were lower than those from *hcp*. As a reference, the *16S rRNA* gene was used in both conditions and *nifH* was only from the bean nodule bacteroids (Appendix A).

We do not know the reasons why pTssA and pHcp and other T6SS genes have different expression levels yet. This, along with the search for other potential promoter regions in the cluster, will be subjects of future investigations focused on the regulation of the system.

### 3.4. Proteins Re78/79: A New Toxin/Immunity Pair

The genes *re78* and *re79* are the nonstructural genes that showed the highest expression both in free living and in symbiosis (Figure 3); they lie between *hcp* and *vgrG* (Figure 1) and are excellent candidates for encoding an E/I pair, that is, for a toxic effector and its cognate immunity protein. In a proteomic analysis of the supernatant of the wt strain and the *hcp* mutant, peptides corresponding to Re78 and Re79 were identified (Appendix A). The number of Peptide Spectral Match (#PSM) of Re79 was similar in the two strains, but the #PSM corresponding to Re78 in the *hcp* mutant strain was only 10% of that in wt (Figure 4a). This indicates that Re78 is dependent on T6SS for its secretion. Considering that the expression of these proteins has very low values in the ReMim1 cells (Figure 3b), *re79* was fused to a 3′ sequence to provide a C-terminal Strep-tag and introduced into plasmid pLMB509 under the control of a inducible taurine promoter, generating plasmid pLMB509-Re79StrepTag. This plasmid was conjugated to strains ReMim1 and hcp::pk18 and cell fractionation was performed from the induced cells. The protein Re79 was identified in the periplasmic fraction from both strains. This is consistent with the presence of a signal peptide identified in its sequence and with a T6SS-independent translocation (Figure 4b).

The data suggest that re78/79 could code for a toxin and immunity protein, respectively, that would act in the periplasmic space. To corroborate this hypothesis, we studied the effect of the expression in *E. coli* of Re78 and Re79 proteins using plasmids pET22b and pBAD33, respectively. The growth of *E. coli* capable of expressing two variants of Re78, one with a N-terminal PelB (Re78PelB) to be directed to the periplasm or without PelB (Re78), and Re79 were compared under induction or repression conditions (Figure 5a,b). The growth of *E. coli* capable of expressing two variants of Re78, one with a N-terminal PelB (Re78PelB) or without PelB (Re78), and Re79 were also compared. Figure 5b shows that, under induction conditions, only the strain expressing Re78PelB had no growth. The rest of the strains, including the control strain (with plasmid pET22b and pBAD33), grew relatively well, although those expressing Re78 and Re78–Re79 were delayed 1–2 h compared to the others. These results indicate that Re78 inhibits growth when it has a signal peptide and that this effect is neutralized by the coexpression of cognate immunity protein Re79. In addition, the microscopic examination was performed three hours after the induction of cells expressing Re78PelB and a control with plasmid pET22b (Figure 5c). It is difficult to quantify the difference between the images obtained in these two cases, but it appears as though there could be more aggregation of cells in the case of *E. coli* expressing Re78PelB, as seen in the insets.

### 3.5. Re78/Re79 Are Involved in Bacterial Competition

To assess whether Re78/Re79 contribute to intrabacterial competition in vitro, a co-incubation assay of the ReMim1 strain and the Δre78–79 strain was performed. The assay showed a significant reduction in the number of cells in the mutant strain in the presence of the wt strain that did not occur when the hcp::pk18 strain and wt were coincubated (Figure 6a). There was also no reduction when the *hcp* mutant was incubated with Δre78–79 (Figure 6b). Similarly, when the growth of the hcp::pk18 or Δre78–79 was compared in the presence of the wt strain using serial decimal dilutions, similar results were obtained, that is, the survival of Δre78–79 was lower and this effect did not occur with the *hcp* mutant co-incubation (Figure 6c). Figure 6d shows that there is no difference in the growth of wt and Δre78–79.

To study if the Δre78–79 mutant is affected in competition for nodule occupancy, a nodulation test was carried out as shown in Figure 7a. To differentiate the strains, they were fluorescently labeled using plasmids pHC60 and pMBC13. Prior to co-inoculation, it was shown that the presence of plasmids does not affect the competitiveness (Figure 7b) and growth (Figure 6d). Additionally, the strains from collected nodules were subsequently confirmed on selective media. The *Phaseolus vulgaris* seedlings were coinoculated with ReMim1 (pHC60) and Δre78–79 (pCMB13) strains at different ratios (Figure 7c). In the case of the 10:1 ratio (ten times more cells than the wt cells), the nodule number did not differ between expected and observed. However, in the 1:1 ratio, only 25% of the nodules were occupied by the mutant strain instead of the 50% expected and, in the 1:10 ratio (ten times more cells of the mutant), only 36% of the nodules were occupied by the mutant strain instead of the 90% expected. These results indicate that the Re78/79 pair contributes to the increased competitiveness of ReMim1 for nodulation.

## 4. Discussion

This study contributes to the understanding of the role of rhizobial T6SS in the rhizosphere and symbiosis. The T6SSs of plant-beneficial bacteria secrete effector proteins that play an important role in communication with their host plant and with the surrounding microbiota [43]. T6SS is present in a variety of rhizobia, and it has been shown that its activity can have a positive, negative, or neutral effect on symbiosis [16]. We showed that mutations in ReMim1 T6SS structural genes have a negative effect in symbiosis with beans [21], thus suggesting that some of the effectors secreted by the system could be responsible for the phenotype. The mutations performed in this work that affect genes with possible effector functions show that they are not responsible for the demonstrated beneficial effect of T6SS in symbiosis. Some mutants induced more nodules, but there was no difference in the shoot dry weight compared to the plants inoculated using the wt strain (Figure 2). This opens the possibility that another effector exists outside the cluster studied or that the structural elements might behave as effectors, as has been proposed for Hcp of the T6SS in *Acidovorax citrulli* [44]. The structural elements are potentially recognized as a signal by the plant, activating an unknown mechanism that results in an effective symbiosis, such as that which occurs with flagellin, the main protein of the bacterial flagella, which is recognized by *Arabidopsis* triggering a defense response that alters its growth [45]. These hypotheses will be investigated in the future.

This led us to analyze whether the role of potential T6SS effectors of ReMim1 is primarily antibacterial as has been described for most of the effectors [10,46]. The bioinformatics analysis and experimental work show this to be true. It was found that the *re82* gene of module 2 could code for a toxic effector with DNase activity as described for the *Agrobacterium tumefaciens* homologous effector, Atu3640, which is T6SS-dependent [40].

The analysis of module 3 showed that four proteins are related to immunity proteins. Three of these contain motifs described in these proteins (GAD, T6SS Tdi1C, and SMI1/KNR4). In addition, Shi and collaborators showed that the closest structural homolog of the GAD-like domain in the Tdi1 of *A. tumefaciens* is SMI1/KNR4 [47]. The other protein has a signal peptide suggesting that its secretion does not depend on the T6SS apparatus as is the case for the effectors. The other two proteins of module 3, Re87 and Re89, are small proteins. The Re87 protein shows an amino acid identity of 88% to a SAM-dependent methyltransferase. Re89 has no homology to database proteins. This suggests that this module would have a primarily defensive role against toxic effectors of competing microorganisms. In multimicrobial environmental populations, acquisition of immunity genes by bacteria, even lacking the corresponding effector, is a strategy to defend against hostile T6SS [48]. The *Vibrio cholerae* strains have orphan immunity genes encoded at the end of T6SS clusters that evolved through recombination of horizontally acquired modules [49]. The 3′ region of ReMim1 T6SS cluster could be a consequence of gene duplication or swapping events in an environment where competing cells or similar strains contain T6SSs [12,49,50].

Regarding module 1, we found no evidence for its function by performing comparative analysis with known effectors or against databases, but the E/I pairs are frequently close to *vgrG* and *hcp* genes [51,52,53,54,55].

Protein 79 has a signal peptide and has been localized in the periplasm. There are numerous examples of E/I pairs in which the immunity protein has a signal peptide to act in the periplasmic space neutralizing its cognate effector acting as peptidoglycan-degrading enzymes (*Pseudomonas aeruginosa* [56,57]; *Agrobacterium tumefaciens* [40]; and *Vibrio parahaemolyticus* [58]) or as phospholipases (*Neisseria cinerea* [59]; and *Pseudomonas aeruginosa* [60]). Re78 had a toxic effect on *E. coli* when a signal peptide was added, and this effect was neutralized in the presence of Re79. In cases of anti-peptidoglycan toxin expression in *E. coli*, cells with altered morphologies have been seen: amorphous, elongated, filamentous, swollen and spherical [29,56,57]. In our case, no such cells were seen, although they appear more aggregated. Re78 has none of the domains described in peptidoglycanases, ion-selective pore-forming proteins, and lipases or in any other toxin [12,29,61,62], so these proteins might constitute a new E/I pair with extracellular action. Re78/79 pair distribution among other bacteria is restricted to a few species mainly of the family *Rhizobiaceae* and other species of Alphaproteobacteria (Figure 1).

Rhizobia dwell in bulk soil, in the rhizosphere, and in nodules. Studies on plant microbiota show that the widely distributed T6SS is an important factor in competition for the natural niche and for plant protection [17,43,63,64,65]

The fact that the Δre78–79 is less competitive in intraspecific co-incubation and nodule occupancy trials indicates the relevance of the Re78/79 pair to overcome competition from other bacteria. Similarly, it has been proposed that the T6SS of legume symbionts such as *P. phymatum* with *A. caulinodans* that efficiently nodulate *V. unguiculata* and *S. rostrata*, respectively, is important in interbacterial competition [19,20]. Hug and collaborators proposed that T6SS-b is relevant in the early stages of the symbiosis of *P. phymatum* with *V. unguiculata* [66]. It has also been demonstrated that the *Vibrio fischeri*, which is a symbiont located in light-organ crypts of the squid *Euprymna scolopes*, uses a T6SS to eliminate competitors from co-occupying sites in its natural host [67]. Nodule communities are formed by diverse endophytes, mainly belonging to the phyla Pseudomonadota, Actinomycetota, and Bacillota [68,69].

Future work should address the importance of T6SS in nodule occupancy and protection in a community context considering that T6SS acts not only against Gram-negative bacteria but also against Gram-positive bacteria [70,71], without forgetting that some effectors could affect eukaryotic hosts, although an effector secreted to plant cells has not yet been identified [15].

In this work, the importance of a novel E/I pair of the T6SS of ReMim1 has been discovered and functions for several genes in the cluster have been proposed, but more research is needed to understand the activity of rhizobial T6SS-dependent effectors and their relevance in interactions with soil, the rhizosphere, nodule bacteria, and their legume hosts.

## 5. Conclusions

Three non-structural gene modules were identified in the T6SS cluster of ReMim1. Two of these modules include Effector–Immunity pairs: re82/83 encoding a DNase and re78/79-encoding presumably a novel periplasmic-acting toxin of unknown function, conserved in some strains mainly from the Rhizobiaceae family that is important in bacterial competition. The third module encodes immune orphan and truncated proteins. None of these effectors had a positive effect in symbiosis with *P. vulgaris*.

## Figures and Tables

**Figure 1 biology-12-00678-f001:**
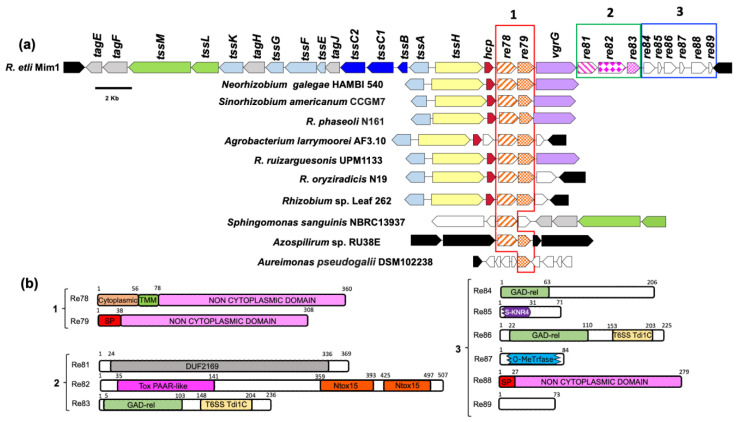
ReMim1 T6SS gene cluster. (**a**) T6SS cluster and conservation of *re78* and *re79* in different bacterial genomic contexts. Orthologous genes have the same color and pattern except for white ones that indicate genes of unknown function and black ones that correspond to genes not related to T6SS. Boxes (modules) 1, 2, and 3 group genes that may encode for effectors. (**b**) Domains identified using the InterProScan server in proteins encoded by modules 1–3. Domains, TMM: transmembrane; SP: Signal peptide; DUF: Domain of unknown function; Tox PAAR-like: PAAR (proline-alanine-alanine-arginine) repeat family; N-Tox15: Novel toxin 15 predicted as endonuclease; GAD-related domain: N-terminal domain of T6SS immunity protein Tdi1 from *Agrobacterium tumefaciens*; T6SS Tdi1: C-terminal domain of the Tdi1 immunity protein; S-KNR4: SMI1/KNR4 domain of immunity proteins in bacterial toxin systems; and O-MeTrfase: domain of O-methyltransferases some of which utilize S-adenosyl methionine as substrate. Numbers above are the domains corresponding to the amino acid position.

**Figure 2 biology-12-00678-f002:**
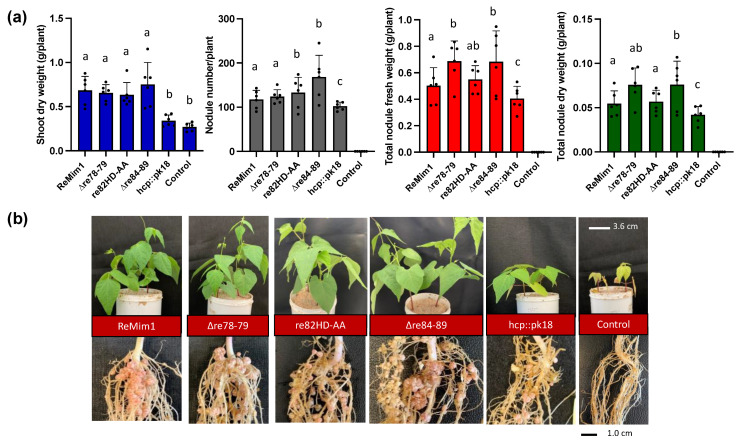
Symbiotic phenotype of *Phaseolus vulgaris* cv. Negro Jamapa 28 dpi with ReMim1 and T6SS mutants. (**a**) Quantitative data from dry weight of plants and nodules. Values are averages of at least six plants from four independent replicates. Each point represents the average of 4 biological replicates. Bars with the same letter are not significantly different using a Student’s *t*-test (*p*  <  0.05). (**b**) Shoot, roots, and nodules’ representatives. Scale bars are 3.6 cm and 1 cm, respectively.

**Figure 3 biology-12-00678-f003:**
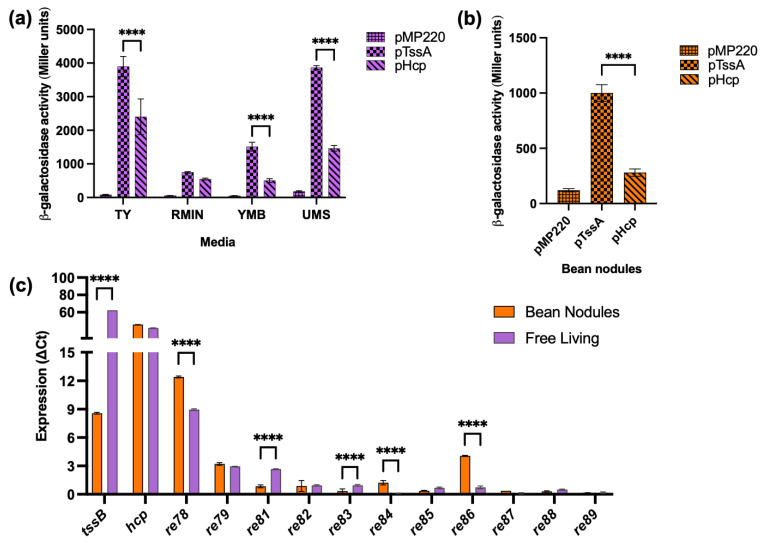
T6SS gene expression. (**a**) Free-living activity of ReMim1 pTssA and pHcp promoters in different media. (**b**) pTssA and pHcp activities from bean bacteroids, pMP220 (empty vector) was used as control. (**c**) RT-qPCR analysis of T6SS genes. Data are mean ± SD, *n* = three biological experiments and three replicates. Statistical significance between samples according to an unpaired two-tailed Student’s *t*-test is denoted by asterisks (**** *p* < 0.0001).

**Figure 4 biology-12-00678-f004:**
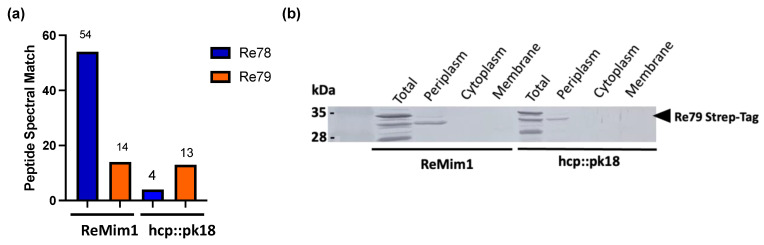
T6SS-dependent Re78 secretion and Re79 subcellular location. (**a**) Peptide spectral match of Re78 and Re79 proteins from ReMim1 and a *hcp* mutant secretomes. (**b**) Identification of Re79(Strep) from ReMim1 and hcp:pk18 cells harboring plasmid pLMB509 Re79-StrepTag. Subcellular fractions were analyzed using Western blot with anti-Strep antibodies. The expected size for Re79StrepTag is ~32 kDa (◀︎).

**Figure 5 biology-12-00678-f005:**
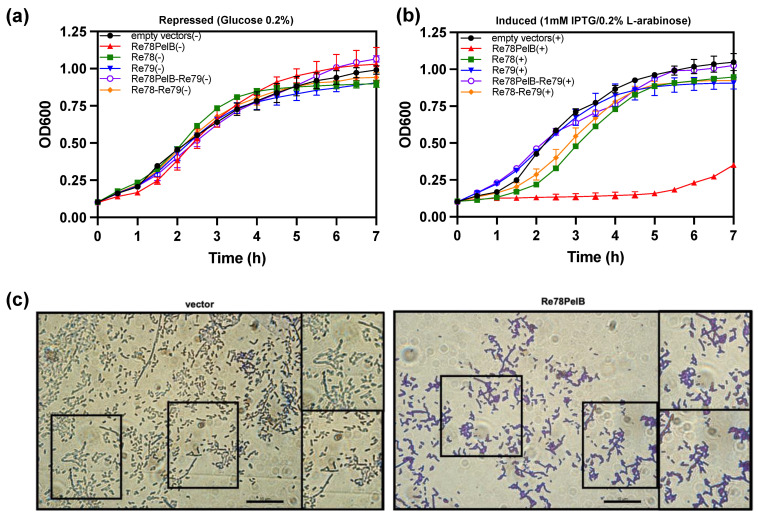
Re78/79 Effector–Immunity pair analysis. Re78 with and without a PelB signal peptide was induced in *E. coli* (BL21DE3) by IPTG 1mM from a pET22b derivative plasmid, while Re79 was induced by arabinose (0.2% *wt./vol.*) from pBAD33. (**a**) *E. coli* growth containing 0.2% glucose (repressed) (−). (**b**) *E. coli* growth (induced) (+). Cultures were incubated during 7 h at 37 °C in a Bioscreen C° Pro system. Data are mean ± SD, *n* = 6 technical replicates from three biological independent experiments. (**c**) Representative images of *E. coli* pET22b (empty vector) and Re78PelB induced 3 h. Cells were stained with Crystal Violet 1% (*wt./vol.*) and visualized (100×) with a Leica DM2000 microscope and a Leica DFC 300FX camera (1.4 Mpixels) and analyzed using ImageJ, scale 10 μm.

**Figure 6 biology-12-00678-f006:**
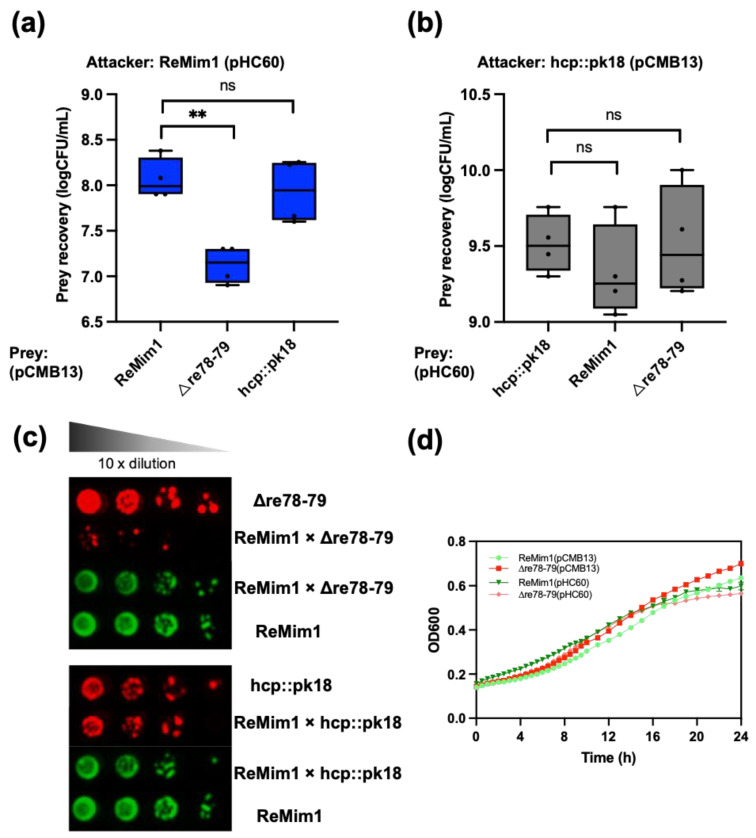
ReMim1 bacterial competition. (**a**) Viability counts of Δre78–79 and hcp::pk18 mutants strains harboring pCMB13 (preys) after 16h in dependent contact at 28 °C in TY plates with ReMim1 pHC60 (attacker) at a 1:1 ratio. (**b**) Viability counts of ReMim1 and Δre78–79 strains harboring pHC60 after 16 h in dependent contact in TY plates with hcp::pk18 mutant pCMB13 (attacker) at a 1:1 ratio. Data are mean ± SD, *n* = four biological experiments and three replicates. Statistical significance between samples according to an unpaired two-tailed Student’s *t*-test is denoted by asterisks (*p* < 0.01); “ns” indicates no statistically significant difference (*p* > 0.05). (**c**) Growth of serial dilutions of ReMim1, Δre78–79, and hcp::pk18 independently and co-incubated as indicated. Media were tetracycline TY for ReMim1 pHC60 and spectinomycin TY for Δre78–79 and hcp:pk18 mutants harboring pCMB13. (**d**) Growth curves of ReMim1 and Δre78–79 with pCMB13 and pHC60 plasmids in TY media at 28 °C used in competition assays. Data are shown as mean ± SD, *n* = five technical replicates.

**Figure 7 biology-12-00678-f007:**
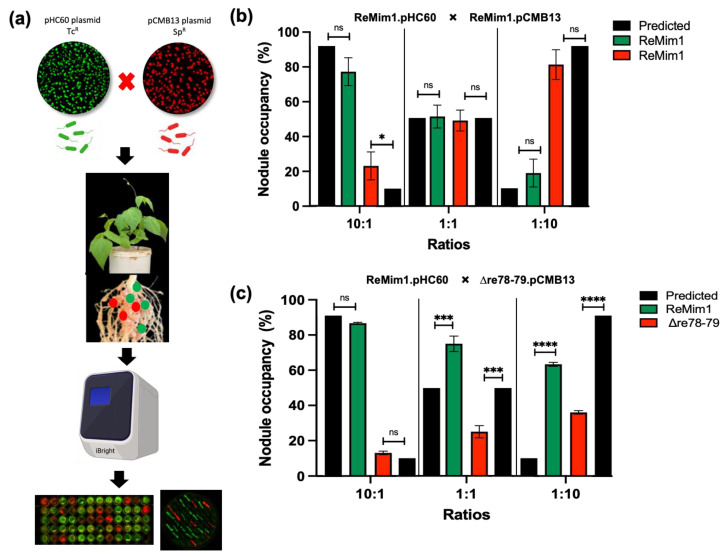
Effect of *re78–79* gene deletion on nodule occupancy competition. (**a**) Experiment workflow for the competition assay among labeled strains with plasmids pHC60 (Tc^r^) and pCMB13 (Sp^r^) able to express fluorescence proteins GFP (green nodules) and (red nodules), DsRED, respectively. Approximately 60–100 nodules per plant were removed at 28 dpi. The identities of the strains from the nodules were ascertained using fluorescence and by growth in YMB medium with the different antibiotics. (**b**) Nodule occupancy test to demonstrate that the presence of plasmids does not affect competitiveness. (**c**) Nodule occupancy after coinoculation with ReMim1 (pHC60) and Δre78–79 (pCMB13) strains at different ratios. Assays were performed in duplicate with two biological replicates. The asterisk denotes significance according to Fisher´s test (****) *p* < 0.0001, (***) *p* < 0.001, (*) *p* < 0.05 and “ns” indicates no statistically significant difference (*p* > 0.05).

## Data Availability

Not applicable.

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
