# Peer review of "The T6SS-Dependent Effector Re78 of *Rhizobium etli* Mim1 Benefits Bacterial Competition"

_biology, 2023, doi:10.3390/biology12050678_

Round 1
Reviewer 1 Report
The research about the role of type VI secretion system in Rhizobium etli discovers the role of some genes of this secretion system during establishing the symbiosis with legumes. It was interesting to read and to know that not only plant regulate the number of nodules but bacteria take part in nodulation process too. However, there are still many questions.
In my opinion the title of the article should be changed. The title of your previous work was “T6SS has a positive effect on symbiosis”, this work titled as “T6SS benefit bacterial competition but not symbiosis”. So were is the truth? In previous work you found that hcp gene is necessary for establishing of symbiosis. In this work you studied some other genes and found that they do not play crucial part in a process of establishing the symbiosis but are necessary for competition among bacteria. Both articles are named as if you studied the whole T6SS, however only some genes of T6SS were studied. Readers will be confused. Maybe to make the title more precise, like “several genes of T6SS …”.
It is better to give a brief review of results with hcp mutant in introduction. Now it is not clear, what is this? why do you mention this? and so on. Moreover in introduction (line 74) it is said that T6SS has a positive role for symbiosis. But you have found a positive role only for one gene. And if T6SS has a positive role on symbiosis, why the title of this article says that T6SS does not benefit for symbiosis. In my opinion. There should be more details on your previous work, and attention should be paid on hcp gene, that is necessary for symbiosis.
Lines 49, 62, 70, 92 -check brackets for references
Lines 63, 77, 252, 309 – check the style of reference
Lines 106, 145, 156, 172, 210, 233, 234, 237, 252 – give the country of the manufacturer
Line 162 – give the reference for Sibinelli-Sousa
Line 178 – what is the time of sterilization
Line 186 – give hours of dark period. I guess it is 8, but it should be written 16/8
Line 207-208 – what is the time of sterilization
Line 333 – 334 - check the sentence
Line 366 – it is better to say “less in bacteroids”
Line 505 – check “that he closest structural homolog” and it is better to say Shi with authors in stead of Shi et al.
Author Response
Dear reviewer, please find the responses in red to all your questions and comments. The lines where changes are indicated correspond to the revised manuscript.
Reviewer 1---------
The research about the role of type VI secretion system in Rhizobium etli discovers the role of some genes of this secretion system during establishing the symbiosis with legumes. It was interesting to read and to know that not only plant regulate the number of nodules but bacteria take part in nodulation process too. However, there are still many questions.
- In my opinion the title of the article should be changed. The title of your previous work was “T6SS has a positive effect on symbiosis”, this work titled as “T6SS benefit bacterial competition but not symbiosis”. So were is the truth? In previous work you found that hcp gene is necessary for establishing of symbiosis. In this work you studied some other genes and found that they do not play crucial part in a process of establishing the symbiosis but are necessary for competition among bacteria. Both articles are named as if you studied the whole T6SS, however only some genes of T6SS were studied. Readers will be confused. Maybe to make the title more precise, like “several genes of T6SS …”.
We agree that the title may be misleading and we modify it. According to the reviewer's suggestion, we specify that effector Re78 is important on bacterial competition.
New title: The T6SS-dependent effector Re78 of Rhizobium etli Mim1 benefits bacterial competition
- It is better to give a brief review of results with hcp mutant in introduction. Now it is not clear, what is this? why do you mention this? and so on. Moreover in introduction (line 74) it is said that T6SS has a positive role for symbiosis. But you have found a positive role only for one gene. And if T6SS has a positive role on symbiosis, why the title of this article says that T6SS does not benefit for symbiosis. In my opinion. There should be more details on your previous work, and attention should be paid on hcp gene, that is necessary for symbiosis.
Our previous work, based on the symbiotic phenotypic analysis of three different structural mutants, indicated that an active T6SS was beneficial for the symbiosis. In the current work we refer to the analysis of the effect of effectors secreted through this system. The ones that we analyze here are not found to be responsible for the observed beneficial effect.
In accordance with this comment, more information from previous work is added on lines 79-82. " Three ReMim1 mutants, affected in T6SS structural genes (hcp::pk18, DtssM and DtssA-tagE), induced bean plant with lower shoot dry weight and smaller nodules than the wild-type strain [21]."
Lines 49, 62, 70, 92 -check brackets for references, corrected
Lines 63, 77, 252, 309 – check the style of reference corrected
Lines 106, 145, 156, 172, 210, 233, 234, 237, 252 – give the country of the manufacturer corrected
Line 162 – give the reference for Sibinelli-Sousa corrected (the reference was at the end of the sentence) (line 168)
Line 178 – what is the time of sterilization.
Information has been added (lines 183 and 212).
Line 186 – give hours of dark period. I guess it is 8, but it should be written 16/8 (line 191).
Line 207-208 – what is the time of sterilization.
Information has been added (lines 183 and 212).
Line 333 – 334 - check the sentence ,
"as mentioned above" has been removed from the text (line 340)
Line 366 – it is better to say “less in bacteroids”
We agree with this change (line 374).
Line 505 – check “that he closest structural homolog” and it is better to say Shi with authors instead of Shi et al.
We modify the sentence "In addition Shi and collaborators showed that the ….(line 521).

Reviewer 2 Report
De Sousa and co have explored the role of two genes located within a cluster of genes which are responsible for the T6SS. The genes code for two new proteins that appear to serve as a toxin and immunity protein pair. Furthermore, the manuscript goes on to show that this toxin/immunity pair play a role in competition between bacteria. The data is interesting and novel. However, the manuscript, in my view, is not ready for publication, as I feel that it would benefit from some changes to help the reader more easily follow the storyline. More effort needs to be made to increase readability.
For the simple summary, it is not clear what this gene pair has to do with the T6SS, apart from the location of the genes within the cluster. They operate independently from the T6SS? If so, the authors should clarify what their connection is to the T6SS.
Line 28, the authors claim that the Re78/Re79 proteins behaved as a toxic effector/immunity pair in E. coli. But the authors don’t show how these proteins help E. coli compete against other bacteria. They only show that E. coli does not like to express Re78 with a PelB tag, i.e., that Re78-PelB is toxic. So can they really claim to have shown that these proteins behaved as a toxic effector/immunity pair? What about an experiment with E. coli versus E. coli expressing the pair, or even better, E. coli versus ReMim1, with and without the pair?
Line 57, it would be helpful to know at this point in the manuscript how many genes are generally involved in the TssA-M and Tag. 20 genes, or 50?
Line 70 Since then, ….. Please rewrite this sentence, as the grammar cannot be correct.
Line 74 Please provide more details to the ‘positive role’, e.g., how big was the effect? Positive role is too vague.
Line 79 How can the authors provide evidence supporting the expression conditions (statement does not make sense)? Uncovering the expression levels under various conditions?
After printing the manuscript, the text of the figures was too small to read. FigS2 was impossible. Please consider a larger font.
Figure 1 Do re78 and re79 genes each have their own specialized promoters? Are they (additionally) transcribed from the hcp promoter? There should be some comments from the authors about this aspect, as this is highly relevant for their interpretations of expression levels, e.g., Fig. 3a&b. I note that the authors have correctly provided expression data (RT-qPCR) to confirm the expression of these genes. However, re78 and re79 appear to have quite different expression levels judging by the RT-qPCR.
Furthermore, where is the promoter for clpV, and does the use of the reporter plasmids (pHcp and pTssA) mean that clpV is overexpressed?
Please add the label tssH to Figure 1. It is difficult to follow the manuscript reference to tssH (line 355) without hectically running back through the manuscript to find where tssH is also referred to as clpV (Line 56).
Line 333 Too many ‘as mentioned’.
Line 338 How is the shoot dry weight higher in the plants inoculated with the three mutants than with the WT? Only one mutant causes higher shoot weight, and this is only slight. The authors need to be more careful!
Line 343 Please provide a sufficient description of the mutant hcp::pk18. This is important, particularly when it is unknown to the reader whether this pK18 insertion ought to knockout both hcp function as well as the expression of downstream genes.
Line 405 Please explain what PelB is upon its first appearance, otherwise the reader is lost.
Line 411 I would interpret this differently to ‘similar growth’, i.e., growth was retarded/delayed by 1-2 hours upon expression of Re78 and Re78-Re79.
Line 416 ‘a greater number of aggregates’ is too vague, i.e., how much greater? And what is an aggregate exactly? A less defined shape’ is also very vague. The authors should provide more clarity here.
Figure 6 The statement for statistical significance falls under (c) but is only relevant to (a) and (b).
Author Response
Dear reviewer, please find the responses in red to all your questions and comments. The lines where changes are indicated correspond to the revised manuscript.
Reviewer 2...........
De Sousa and co have explored the role of two genes located within a cluster of genes which are responsible for the T6SS. The genes code for two new proteins that appear to serve as a toxin and immunity protein pair. Furthermore, the manuscript goes on to show that this toxin/immunity pair play a role in competition between bacteria. The data is interesting and novel. However, the manuscript, in my view, is not ready for publication, as I feel that it would benefit from some changes to help the reader more easily follow the storyline. More effort needs to be made to increase readability.
- For the simple summary, it is not clear what this gene pair has to do with the T6SS, apart from the location of the genes within the cluster. They operate independently from the T6SS? If so, the authors should clarify what their connection is to the T6SS.
To clarify this issue, paragraphs from lines 14-17 have been modified....
The T6SS is a nanosyringe able to secrete proteins called effectors to both eukaryotic and prokaryotic target cells. The ReMim1 T6SS gene cluster encodes for a toxic effector (Re78) together with an immunity protein (Re79) as demonstrated when expressed in Escherichia coli.
Line 28, the authors claim that the Re78/Re79 proteins behaved as a toxic effector/immunity pair in E. coli. But the authors don’t show how these proteins help E. coli compete against other bacteria. They only show that E. coli does not like to express Re78 with a PelB tag, i.e., that Re78-PelB is toxic. So can they really claim to have shown that these proteins behaved as a toxic effector/immunity pair? What about an experiment with E. coli versus E. coli expressing the pair, or even better, E. coli versus ReMim1, with and without the pair?
Many T6SS-dependent effectors have a toxic effect when injected into the target cell. To avoid an autointoxication of the toxin-producing bacteria, in the T6SS gene cluster, next to the gene encoding the toxin, in many cases there is another gene encoding a so-called immunity protein that allows neutralization of the toxin. In our work, we have found that Re79 has a functional signal peptide, and this is usually associated with immunity proteins against toxins that act in the periplasm. So, we think that Re78 could be an effector secreted by the ReMim1 T6SS nanosyringe in the periplasm of the target cell. When expressing those proteins in E. coli we see that Re78 has a toxic effect only when the leader peptide PelB send the protein to the periplasm. This does not occur when Re79 is present, that is why we consider Re78/Re79 a toxic effector/immunity protein pair.
To make the text clearer and considering the space limitation, it has been modified between lines 28 and 30.
- Line 57, it would be helpful to know at this point in the manuscript how many genes are generally involved in the TssA-M and Tag. 20 genes, or 50?
This information has been added, line 58 "often comprising 20-30 genes "
- Line 70 Since then, ….. Please rewrite this sentence, as the grammar cannot be correct.
The line has been corrected, " Later on, it was observed that the presence of T6SS does not affect the symbiotic effectiveness of..." (Lines 70-71)
- Line 74 Please provide more details to the ‘positive role’, e.g., how big was the effect? Positive role is too vague.
More details have been provided in lines 75 and 77 "; in both cases, mutants in structural genes of the T6SSs induced plants with up to 40% less shoot dry weight than wild type, and up to 60% less nodule fresh weight [21,22].
- Line 79 How can the authors provide evidence supporting the expression conditions (statement does not make sense)? Uncovering the expression levels under various conditions?
The sentence has been rephrased to indicate the conditions "under free-living conditions and symbiosis with beans by transcriptional fusions of the T6SS promoter region and by RTq-PCR” (lines 84-85).
- After printing the manuscript, the text of the figures was too small to read. FigS2 was impossible. Please consider a larger font.
We have modified the figure by including a larger font and the figures have been made larger.
- Figure 1 Do re78 and re79 genes each have their own specialized promoters? Are they (additionally) transcribed from the hcp promoter? There should be some comments from the authors about this aspect, as this is highly relevant for their interpretations of expression levels, e.g., Fig. 3a&b. I note that the authors have correctly provided expression data (RT-qPCR) to confirm the expression of these genes. However, re78 and re79 appear to have quite different expression levels judging by the RT-qPCR.
The questions raised are very interesting, but we do not yet have the answers. Since the separation of these ORFs is only 11 nucleotides, we do not expect different promoters for re78 and re79. We are not sure about the reasons for the difference in the expression levels of re78 and re79, we believe that they might be transcribed in the same mRNA, but there might be differences in the stability of longer transcripts. Regarding whether they can be transcribed with hcp, there are 146 nucleotides between hcp and re78 so independent transcription cannot be ruled out.
We include these considerations in a paragraph in the lines 379-381.
- Furthermore, where is the promoter for clpV, and does the use of the reporter plasmids (pHcp and pTssA) mean that clpV is overexpressed?
What we have called pHcp contains the intergenic region between tssA and tssH (clpV) line 221. This region should contain the promoter of tssH and we think that it would also be the same as that of hcp, the separation of tssH and hcp is only 41 nucleotides, although we must investigate it as we commented in the previous answer. The pHcp fusion contains only the intergenic region between tssA and tssH and not the complete tssH gene so this gene would not be overexpressed.
Please add the label tssH to Figure 1. It is difficult to follow the manuscript reference to tssH (line 355) without hectically running back through the manuscript to find where tssH is also referred to as clpV (Line 56).
We apologize for the confusion, in order to make this clearer we have labeled tssH in the revised Figure 1.
Line 333 Too many ‘as mentioned’. Corrected (line 340).
Line 338 How is the shoot dry weight higher in the plants inoculated with the three mutants than with the WT? Only one mutant causes higher shoot weight, and this is only slight. The authors need to be more careful!
We thank the reviewer for catching this mistake, and we have corrected the text in lines 346-348.
- Line 343 Please provide a sufficient description of the mutant hcp::pk18. This is important, particularly when it is unknown to the reader whether this pK18 insertion ought to knockout both hcp function as well as the expression of downstream genes.
We think this is an important clarification and we have added the information that the mutation was not polar in line 348 (we show this by mutant complementation described in reference 21).
- Line 405 Please explain what PelB is upon its first appearance, otherwise the reader is lost.
We have added an explanation upon first mention of PelB (line 132 “It has to be noted that the PelB leader peptide from pET22b, N-terminally attached to a protein, directs it to E. coli periplasm” and on line 418 “to be directed to the periplasm”.
- Line 411 I would interpret this differently to ‘similar growth’, i.e., growth was retarded/delayed by 1-2 hours upon expression of Re78 and Re78-Re79.
Your assessment is more accurate and the lines 423-424 have the changes:” although those expressing Re78 and Re78-Re79 were delayed 1-2 hours compared to the others”.
Line 416 ‘a greater number of aggregates’ is too vague, i.e., how much greater? And what is an aggregate exactly? A less defined shape’ is also very vague. The authors should provide more clarity here.
It is true and the last part of the sentence in line 428 has been deleted.
It is true that these statements made in the manuscript are somewhat vague. Indeed, we ourselves lack a concrete idea of how to quantify the “shape” of the cells, or exactly how to define an aggregate, so it is difficult to attach exact numbers to these statements. Since we would like to avoid any confusion or misinterpretation, we have therefore chosen to remove any firm claims from the manuscript. Instead on line 428, we write “It is difficult to quantify the difference between the images obtained in these two cases, however it appears as though there could be more aggregation of cells in the case of E.coli expressing Re78PelB, as seen in the insets.”
- pointing out the difficulty in measuring the differences but still drawing the reader’s attention to the possibility of an effect.
Figure 6 The statement for statistical significance falls under (c) but is only relevant to (a) and (b). Corrected

Round 2
Reviewer 2 Report
Please check Figure 4b.
Author Response
Dear reviewer,
Please find enclosed our revised manuscript (Round 2).
We have modified Fig.4 to avoid the PDF version being defective as detected.
